# Improving the Physical and Mechanical Properties of Mycelium-Based Green Composites Using Paper Waste

**DOI:** 10.3390/polym16020262

**Published:** 2024-01-17

**Authors:** Thana Teeraphantuvat, Kritsana Jatuwong, Praween Jinanukul, Wandee Thamjaree, Saisamorn Lumyong, Worawoot Aiduang

**Affiliations:** 1Northfield Mount Hermon School, Mount Hermon, MA 01354, USA; kennythana@gmail.com; 2Office of Research Administration, Chiang Mai University, Chiang Mai 50200, Thailand; kritsana.ja@cmu.ac.th; 3Department of Biology, Faculty of Science, Chiang Mai University, Chiang Mai 50200, Thailand; 4Faculty of Architecture, Chiang Mai University, Chiang Mai 50200, Thailand; praween_ji@cmu.ac.th; 5Department of Physics and Materials Science, Faculty of Science, Chiang Mai University, Chiang Mai 50200, Thailand; wandee.th@cmu.ac.th; 6Center of Excellence in Microbial Diversity and Sustainable Utilization, Chiang Mai University, Chiang Mai 50200, Thailand; 7Academy of Science, The Royal Society of Thailand, Bangkok 10300, Thailand

**Keywords:** green composite materials, mycelium technology, agro-industrial waste management, bio-fabrication, SDG 7

## Abstract

The growing demand for environmentally friendly and sustainable materials has led to the invention of innovative solutions aiming to reduce negative impacts on the environment. Mycelium-based green composites (MBCs) have become an alternative to traditional materials due to their biodegradability and various potential uses. Although MBCs are accepted as modern materials, there are concerns related to some of their physical and mechanical properties that might have limitations when they are used. This study investigates the effects of using paper waste to improve MBC properties. In this study, we investigated the physical and mechanical properties of MBCs produced from lignocellulosic materials (corn husk and sawdust) and mushroom mycelia of the genus *Lentinus sajor-caju* TBRC 6266, with varying amounts of paper waste added. Adding paper waste increases the density of MBCs. Incorporating 20% paper waste into corn husks led to the enhancement of the compression, bending, and impact strength of MBCs by over 20%. Additionally, it was also found that the MBCs produced from corn husk and 10% paper waste could help in reducing the amount of water absorbed into the material. Adding paper waste to sawdust did not improve MBC properties. At the same time, some properties of MBCs, such as low tensile strength and high shrinkage, might need to be further improved in the future to unlock their full potential, for which there are many interesting approaches. Moreover, the research findings presented in this publication provide a wealth of insightful information on the possibility of using paper waste to improve MBC performance and expand their suitability for a range of applications in sustainable packaging materials and various home decorative items. This innovative approach not only promotes the efficient utilization of lignocellulosic biomass but also contributes to the development of environmentally friendly and biodegradable alternatives to traditional materials.

## 1. Introduction

In recent years, the growing attention and demand for environmentally friendly products, along with the need to reduce carbon emissions, microplastic pollution, and plastic contamination in the world ecosystem, has led to the development of biodegradable material innovations across a wide range of industries [1]. Within the technological possibilities investigated, manufacturing mycelium-based green composites (MBCs) has emerged as an idea of great interest and a possible alternative for the current development of environmentally friendly materials [2]. MBCs are derived by cultivating the vegetative portion of mushroom mycelia on various lignocellulosic substrates, which act as a biological binder that bonds substrate particles through a biotechnological process. This process holds significant potential for developing a versatile and biodegradable material, offering an alternative to traditional synthetic and other non-renewable materials [3,4]. MBCs offer the advantage of displaying a diverse range of properties that vary based on production factors such as substrate type, fungal species, and various techniques employed during the manufacturing process [5]. Nowadays, MBCs are receiving more attention in academic and commercial research due to their low energy consumption during growth, zero byproducts, and wide range of potential applications [5,6]. While MBCs are generally accepted [3,5] and lauded for their eco-friendliness, some of their mechanical and physical properties require further enhancement to fully realize their potential in various applications [7,8].

The physical and mechanical properties of materials are essential parts in determining their suitability for various applications [9]. Even though MBCs have the advantages of being renewable and biodegradable, they still lack some strength, durability, and other essential properties when compared to conventional materials [10,11]. Addressing these limitations is crucial to unlocking the full potential of MBCs and promoting their wide applications across industries. To address these drawbacks and unlock the full potential of MBCs, many researchers and innovators have been inventing various approaches for improving the properties of MBCs [12,13,14]. Previously, the properties of MBCs were improved by applying a suitable substrate type, fungal species, and fabrication technology [15]. Another possible approach for improving the properties of MBCs is mixing various kinds of agricultural substrates during the production process, resulting in the opportunity to effectively change the properties of various aspects of materials [16]. Moreover, many earlier studies regarding MBC production have also used silica and carbonate sand along with clay to improve the properties of this material [17,18,19,20,21]. In this context, paper waste, which constitutes a significant portion of global solid waste, has become an attractive choice for improving the properties of MBCs because paper waste typically contains cellulose (82–95 wt%), which has an excellent ability to improve the mechanical and physical properties of composite materials [22,23]. Several previous studies have utilized paper waste to improve the qualities of various composite materials because cellulose fibers are known for their high strength and stiffness. When these cellulose fibers are mixed into composite materials, they can reinforce the matrix material [24,25,26,27].

This research delves into the interesting possibility of improving the physical and mechanical properties of mycelium-based green composites through the integration of paper waste. Through a comprehensive investigation, we aim to illustrate that this technique can generate MBCs that are greater in strength, durability, and appropriate for a wider range of applications. Additionally, this research not only contributes to the deployment of environmentally friendly materials but also supports the longer-term goals of promoting circular economy principles and reducing the amount of waste from agricultural sectors.

## 2. Materials and Methods

### 2.1. Source of Substrate and Preparation Processes

Two different types of agricultural waste, namely corn husk and rubber wood sawdust, were used as the main substrates in this experiment. Both substrate types were sourced from agricultural areas and sawmills located in Chiang Mai, Thailand. Before being employed in testing, corn husks were ground in a grinder to make the particles smaller. Similarly, rubber wood sawdust was obtained by cutting wood into pieces using an electric saw during the wood processing phase. After that, rubber wood sawdust and corn husk were sieved through 10 mm sieves. Particles in a size range of 1–10 mm were collected and used for testing. For paper waste, reusable paper was sourced from offices and printing services within Chiang Mai University, Thailand. Before being utilized in the experiment, the paper was soaked in water for five min and then ground in a blender. The humidity of blended paper was then reduced to approximately 60% by pressing out the water and measured using a moisture meter. The paper used was autoclaved for 15 min at 121 °C before being used as an ingredient in molding.

### 2.2. Sources of Mushroom Mycelia and Culture Conditions

A pure culture of mushroom mycelia in the genus *Lentinus sajor-caju* TBRC 6266 used in this study was sourced from the culture collections of the Thailand Bioresource Research Center (TBRC), Thailand Science Park, Thailand. This species has been previously studied and reported to exhibit the potential for producing MBCs with several advantageous characteristics [28]. Pure mycelium was grown on potato dextrose agar (PDA; Conda, Madrid, Spain) and incubated at 30 °C for 7 days. 

### 2.3. Mold Designs and Mycelium-Based Green Composite Production

#### 2.3.1. Mold Preparation and Sterilization

In this study, clear acrylic sheets and cylindrical acrylic were used to create molds for the manufacturing of MBCs. The molds used in molding samples for the flexural, impact, tensile, and water absorption tests were designed using clear acrylic sheets and cut into rectangular shapes (150 × 12.7 × 10 mm and 63.5 × 12.7 × 12.7 mm), a dumbbell-shaped segment (165 × 19 mm, neck 57 × 13 mm), and a square shape (50 × 50 × 10 mm), respectively. These dimensions adhere to the standard methods outlined by the American Society for Testing and Materials (ASTM), including ASTM D790, ASTM D256, ASTM D638, and ASTM D570 [28,29,30,31,32,33]. The molds used in molding samples for the compression test were made from clear cylindrical acrylic with dimensions of 64 mm in diameter and 80 mm in height, which were modified in accordance with the method from Elsacker et al. [34]. Before being used for molding, all molds were sterilized by soaking in a 2% solution of sodium hypochlorite (sodium hypochlorite/hichlor; Chiang Mai, Thailand) for 5 min, followed by two rinses with sterile distilled water [35].

#### 2.3.2. Inoculum Preparation

Sorghum grain was employed to prepare the mycelial inoculum for this experiment. The sorghum grains were cleaned before being boiled for 20 min. After cooling, 100 g of the boiled grains were placed in glass bottles, then covered with a cotton wool plug and autoclaved at 121 °C for 20 min. The bottles were then given 24 h to cool to the ambient temperature. Subsequently, mycelial plugs of approximately 1 × 1 cm obtained from colonies cultured on PDA were transferred into bottles afterward (5 plugs per bottle). The transferred bottles were then incubated at 30 °C for 2 weeks in complete darkness until the sorghum grains became completely covered by mushroom mycelia [28].

#### 2.3.3. Preparation of the Substrate for Mycelial Growing

Each agricultural waste was mixed with the nutritional supplement for mushroom growing (5% rice bran, 1% calcium carbonate, 2% calcium sulfate, and 0.2% sodium sulfate) [36]. The prepared mixtures were then adjusted for their overall moisture content by adding water and measuring using a moisture meter to obtain an optimal level of 60%. One kilogram of mixed substrates was placed into polypropylene bags with a size of 0.09 m wide and 0.32 m long, packed using polyvinyl chloride bag covers that contain cotton plugs, and autoclaved at 121 °C for 60 min. After allowing each substrate to cool down to ambient temperature for 24 h, five grams of mycelial inoculum was then inoculated on top of the substrate in each bag. The bag cultures were subsequently cultured at 30 °C in dark conditions for 21–30 days, at which point the mushroom mycelia completely covered the substrate [28].

#### 2.3.4. Green Composite Molding

The fully colonized substrates, with a humidity level of approximately 60–67%, were ground again and mixed with prepared paper waste in varying ratios: 100:0%, 90:10%, 80:20%, 70:30%, and 60:40% (wt/wt), placed into a designed mold and cold-pressed for 10 min at 0.5 MPa on a unidirectional press machine (Shop press ZX0901E-1, New Taipei, Taiwan). All the molds were incubated at 30 °C in dark conditions for 3 days. After completing three days of incubation, MBC samples were removed from the mold and incubated for another 3 days in a plastic box, allowing the mushroom mycelia to cover all the surfaces. The obtained MBCs were subsequently dried in an oven at 70 °C for 24–48 h to the point at which their mass had become stabilized [28]. After drying, the MBCs were stored in desiccators for maintenance of their humidity levels below 10% for further investigation.

### 2.4. Determination of Physical Properties

#### 2.4.1. Density

After drying, the density of the MBC samples was measured from the test piece used in the compression testing, employing the International Organization for Standardization (ISO) 9427 [37] formula, by calculating the mass of the MBCs divided by their volume. Each treatment was evaluated using ten replications.

#### 2.4.2. Shrinkage

The shrinkage percentage of the MBC samples was measured and calculated using the standard formula described by Aiduang et al. [28] according to their volumes before drying and the volumes of the specimen pieces utilized in the compression testing after drying. The following formula was used to calculate shrinkage: Shrinkage rate (%) = (V_1_ − V_2_/V_1_) × 100, (where V_1_ is volume prior to drying and V_2_ is volume after drying). Ten replications were performed to evaluate each treatment.

#### 2.4.3. Water Absorption and Volumetric Swelling

Water uptake and volumetric swelling of the produced MBCs were evaluated according to ASTM D1037-12 [33] using 10 replications for each treatment. Before testing, the MBC samples were dried at 70 °C until their mass became stable, at which point the initial weight was measured and they were allowed to cool down in a desiccator for a period of 24 h. After that, MBC samples were immersed in deionized water for a total of 24 h. At 2, 4, 6, 12, 16, and 24 h, samples were weighed. The increase in weight was calculated using the formula as follows: Weight increase (%) = [(W − D)/D] × 100 (where W is wet weight, and D is dry weight). Volumetric swelling of the produced MBCs was determined after immersing the samples in distilled water for 2 and 24 h. Afterward, the volumetric swelling values were calculated using the volume change in comparison to the initial volume [38].

### 2.5. Determination of Mechanical Properties

#### 2.5.1. Compression Strength

The compression strength of MBCs in this study was tested following ASTM D 3501-05 [30,39] using a universal testing machine (Hounsfield-H10Ks, New York, NY, USA) load bench with a 10 kN capacity and a 1 kN load cell under normal conditions. The experiments were run at a controlled displacement duration of 5 mm/min. The load–displacement curve was transformed into a stress–strain curve using the standard formulas for calculating compression stress (σ) and strain (ε), which are detailed in Aiduang et al. [28]. For this experiment, ten replications for each treatment were investigated, with the outcome reported in kPa units.

#### 2.5.2. Flexural Strength

Flexural testing was conducted following ASTM D 790-10 [34]. The investigation was performed in a three-point bending configuration under normal conditions using a universal testing apparatus (Hounsfield-H10Ks, New York, NY, USA) at a cross-head speed of 2 mm/min and clamp support distance of 40 mm. Ten replications were performed for each of the treatments for data analysis to determine flexural strength.

#### 2.5.3. Impact Strength

The Charpy impact test was used to determine impact strength in accordance with the preferred method described in ASTM D-256-10 [28]. Before testing, the MBC samples were placed into the machine and exposed to the swinging pendulum until the samples broke. The energy required to fracture the composite materials was measured and used to estimate their toughness and yield strength. Each of the treatments was replicated a total of ten times. The impact strength values were determined by calculating the energy of the cross-sectional area of the MBC samples using the following formula: I (kJ/m^2^) = K/A [where K is the energy required to fracture the sample (kJ) and A is cross-sectional area (m^2^)].

#### 2.5.4. Tensile Strength

The tensile test was conducted with a universal testing machine (Hounsfield-H10Ks, New York, NY, USA) under normal conditions. The test was conducted in accordance with ASTM D 638-14 using an elongation rate of 2 mm/min and a maximum force of 1 kN [40]. Ten replications were employed for each treatment, and the data were analyzed to generate a stress–strain plot and determine the tensile strength.

### 2.6. Statistical Analysis

The statistics of all experiments using the 10 replications were analyzed via one-way analysis of variance (ANOVA) employing the SPSS program version 17.0; SPSS Inc.; Chicago, IL, USA for Windows. Duncan’s multiple range test was then used to indicate any significant differences (*p* ≤ 0.05) between the mean values.

## 3. Results and Discussion

### 3.1. Determination of Physical Properties

#### 3.1.1. Density

Density is one of the primary physical properties of MBCs and may be an important indicator of the material’s advantageous qualities [8]. Figure 1A shows the density of the MBCs in this study. The results showed that the obtained density value varied according to the type of substrates used. The obtained value ranges from 251.15–322.73 kg/m^3^. The MBCs produced from sawdust (300.55–322.73 kg/m^3^) had a higher density than MBCs made from corn husk (251.15–274.51 kg/m^3^). Moreover, the findings of this investigation indicated that the increase in MBC density varied according to the amount of paper waste added. The MBCs with the highest density (322.73 kg/m^3^) were produced from sawdust with 40% paper waste, followed by MBCs made from sawdust with 30% (318.86 kg/m^3^) and 20% (312.09 kg/m^3^) paper waste, respectively, all of which exhibited a statistically significant difference compared to the baseline of 0% (300.55 kg/m^3^). MBCs created with corn husk had a lower density than those made with sawdust; MBCs made from corn husk and 40% paper waste had the highest density (274.51 kg/m^3^) among the corn husk MBCs, followed by 30% paper waste, 20% paper waste, and 10% paper waste, respectively. Corn husk with 0% paper waste made up the MBC with the lowest density (251.15 kg/m^3^), displaying a statistically significant difference compared to MBCs made from corn husk with the addition of 30% and 40% paper waste. However, the addition of paper waste to corn husks contributed to increased density in MBCs when compared to earlier research by Aiduang et al. [28], which reported MBC density values for corn husks in the range of 220.74–240.99 kg/m^3^. This increase in density could potentially serve as a positive indicator for other physical and mechanical properties of these corn husk MBCs. The results of this study are consistent with numerous research studies on composite materials that indicate that several variables, including substrate type, substrate particles, density of individual materials, volume fraction, porosity, manufacturing process, fiber orientation and layup, filler materials, and fiber/matrix interface, affect the density of various composites [41,42,43,44,45]. The density of MBCs is influenced by the type of substrate utilized, the species of mushroom mycelium, and the pressing technique used during manufacturing [8,46]. Moreover, the outcomes also demonstrated that adding paper waste into the substrate can help to improve MBC density. Importantly, the weight of the finished composites increased slightly when compared to the composites without the addition of paper waste. This outcome is in accordance with previous studies that indicated that paper waste provides a good filler material for improving the properties of composite materials, particularly when the density attributes need to be improved [22,23,27,47].

The measured density values were similar to those of MBCs reported in previous studies, which measured between 25–954 kg/m^3^ [8,43,48]. Interestingly, it was discovered that their density was comparable to that of many synthetic foams (11–920 kg/m^3^), some wood composites (170–780 kg/m^3^) [28,49], as well as paper-based materials (170–800 kg/m^3^) [50,51,52], which suggests the possibility of utilization in a variety of applications.

#### 3.1.2. Shrinkage

In this study, MBCs containing sawdust and 0% paper waste had the lowest degree of shrinkage (6.19%), whereas the MBCs containing corn husk and 40% paper waste had the highest degree of shrinkage (14.71%). Adding 40% paper waste to sawdust resulted in an 8.21% shrinkage. The MBCs made from sawdust had a lower shrinkage rate than those made from corn husk (Figure 1B). As the amount of paper waste added to the substrate increased, the degree of shrinkage also increased. The addition of 10%, 20%, and 30% paper waste to sawdust MBCs led to increased shrinkage values of 6.74%, 7.76%, and 7.94%, respectively. Similarly, corn husk MBCs exhibited increased shrinkage values of 11.27%, 11.64%, and 13.12% when adding 10%, 20%, and 30% paper waste, respectively. However, the observed increases in shrinkage values at these levels in both substrates were not statistically different from those without paper waste. These results align with several previous studies where the percent of shrinkage of materials normally varies according to the type of substrate produced, polymer type, moisture content, filler and additives, manufacturing process, and drying temperature [53,54]. The shrinkage percentage of the MBCs was influenced by the substrate used and the initial moisture content within the composites [28,34,55].

The degree of MBC shrinkage in this investigation may be influenced by filler and additives. Normally, paper waste added into a substrate before molding often contains moisture. After it is added to a composite material and the molding process is completed, this moisture can be released during the drying process. The loss of moisture may lead to voids forming inside the composites, which contributes to the loss of volume of material and causes shrinkage [56]. Moreover, the shrinkage values of MBCs in this current study were within the ranges of those in previous studies—between 6.20–16.31% [28,34]. Moreover, the obtained values were also comparable with those of various wood-based composite materials (0.3–30.28%) and some paper-based products (1–12%), yet it was higher when compared with those of materials made of synthetic foam (0.2–2.5%) [28,49,57,58]. Nevertheless, the information obtained might be useful in designing molds to reduce and solve problems regarding material shrinkage.

#### 3.1.3. Water Absorption and Volumetric Swelling

The water absorption test and volumetric swelling assessment results of the MBCs obtained in this study are shown in Figure 2. The findings demonstrated that the water absorption capacity varied according to the type of substrate used and the amount of paper waste added. MBCs produced from corn husk (123.46–166.28%) exhibited a lower water absorption rate than those produced from sawdust (174.50–197.15%) when immersed in water for 24 h (Figure 2A,B). The water absorption of the corn husk MBCs increased sharply during the first 6 h and slowly stabilized after 12 h. Similarly, water absorption of the sawdust MBCs grew sharply in the first 2 h but slowly stabilized after 4 h. The results of this study showed that MBCs produced from corn husk containing 10% paper waste (123.46%) exhibit lower water absorption compared to MBCs without paper waste (132.44%). Adding paper waste to sawdust caused MBCs to have higher water absorption according to the amount of paper waste added. Moreover, the water absorption ability of MBCs produced in this investigation was found to be within the ranges reported in earlier studies, which ranged from 24–560% after immersion in water for a duration of 24–192 h [8,34,40,59].

Primarily, the water absorption capacity of MBC materials was influenced by a wide range of factors, including mycelium species, substrate type, substrate composition, hydrophobic coatings, density, and porosity [18,28,34,49,59,60]. In this study, it was found that differences in the water absorption capacity of MBC materials may be influenced by hydrophobic coatings and substrate composition. Several studies have suggested that the coating of hydrophobic mycelium on substrates used in the manufacturing of MBCs at high densities leads to a decrease in the water absorption capacity of the material [40,49,61,62]. This investigation showed that the MBC samples produced from corn husk had a better covering of hydrophobic mycelium on substrates than the samples produced from sawdust (Figure 3), which led to lower water absorption capacity. The amount of increased water absorption based on the level of paper waste added may be influenced by the cellulose element in the substrate mixture, which contains a large number of accessible hydroxyl groups that contribute to more water absorption [28]. Moreover, low water absorption of MBCs made from corn husk with 10% paper waste added increased the density, and adding paper waste to the substrate mixture in that amount might not have any effect on the hydrophobic mycelium coating.

Although MBC materials still exhibit higher water absorption than conventional materials like synthetic foams and some wood-based products, using this material for interior decoration may not have any effect on its properties. Regardless, the water absorption capacity of MBCs may be advantageous when used as cushioning material or packaging material for liquid chemical products that need to be absorbed in the case of a leak during transportation.

Volumetric swelling serves as an indicator of the stable dimensional performance of composite materials [63]. In this study, the MBC volumetric swelling is shown in Figure 2C. It was discovered that the MBCs exhibited similar swelling after 2 h, which ranged from 1.06–5.13% for MBCs produced from corn husk and 1.65–5.61% for MBCs produced from sawdust. After 24 h, the swelling of MBCs produced from corn husks grew higher than that of MBCs produced from sawdust, with levels ranging from 4.51–9.41% and 3.18–8.07%, respectively. Commonly, the volumetric swelling of MBCs is directly correlated with the water absorption capability of the material, which leads to changes in volume and dimensions [38]. Thus, the outcomes of this investigation are consistent with those of a prior study, which indicated that the volumetric swelling of MBC materials ranged between 0.28 and 21% [38,62,64].

### 3.2. Determination of Mechanical Properties

#### 3.2.1. Compression Strength

The MBC samples for the compression tests that were obtained in this study are shown in Figure 4. The results of this study found that the compression strength of the MBCs varied according to the different substrates used and the amount of paper waste in each MBC (Figure 5A). The MBCs made from sawdust generally resulted in a higher degree of compression strength than those made from corn husk. When paper waste was added into corn husk MBCs, the compression strength increased by more than 40% from 748.79 kPa with 0% paper waste to 1314.75 kPa with 30% paper waste, showcasing statistically significant differences. After another 10% of paper waste was added to the corn husk MBC, the compression strength decreased to 913.31 kPa. On the other hand, when paper waste was added to sawdust MBCs, the compression strength decreased from 1207.38 kPa with 0% paper waste to 241.58 kPa with 40% paper waste. The outcomes of this study clearly demonstrated that adding paper waste into the substrate influenced the change in compression strength of MBCs. Notably, adding 30% paper waste into corn husks results in an improvement of compression strength (1314.75 kPa) by more than 33% compared to the values reported in prior research by Aiduang et al. [28]. In their study, the compression strength of MBCs from corn husks and *L. sajor-caju* was approximately 870 kPa. The obtained results aligned with the outcomes of many previous studies that found that paper waste influenced compression strength in many types of composite materials [26,27,65]. In this study, it was discovered that adding paper waste into corn husk gave the obtained MBCs a greater strength. This is because adding paper waste to the substrate increases the amount of cellulose fibers, which is a substance with high strength and stiffness that enables the reinforcement of matrix materials [24,25,26,27,66]. At the same time, the compression strength of MBCs made from sawdust was decreased as a result of the smaller mycelial growth along with delayed colonization on the substrate (Figure 4). This may be associated with the high cellulose content in the mixture of substrates used, since cellulose has a complex structure that makes it difficult to break down by fungal mycelia [62,67]. In general, it was discovered that sawdust had a higher amount of cellulose (40–44% dry mass basis) [68] than corn husk (29.3–35.3% dry mass basis) [69,70]. When paper waste was added to the mixture, the amount of cellulose fibers in sawdust increased, which made it more difficult for fungal mycelia to break down and slow down its growth. Additionally, the varying amounts of other chemical elements in the starting materials—in particular, hemicellulose content (29–31% in sawdust and 31–37% in corn husk) and lignin content (28–29% in sawdust and 8–14% in corn husk) [28]—may also influence mycelium growth. Thus, adding paper waste to an appropriate level was crucial in improving the compression strength property of MBCs.

Several earlier studies discovered that the compression strength of MBCs can be impacted by a wide range of factors. Typically, the compression strength of MBCs is primarily influenced by the type of substrate, different fungal species, and pressing techniques used throughout the production process [4,8,34,43,48]. Nevertheless, the findings of our investigation demonstrated that the measured compression strength values were within the range of values provided by previous studies: approximately 13.81–4440 kPa [8,34,43,48,71,72,73]. When compared to conventional materials, it was discovered that the obtained MBCs had values similar to those of many types of synthetic foam (2–482,900 kPa) along with some wood-based (100–25,000 kPa) and paper-based products (40–10,000 kPa) [28,49,52,74], which makes them potentially useful in applications for packaging and home decorative items.

#### 3.2.2. Flexural Strength

The results of the flexural strength test are shown in Figure 5B, where MBCs made from corn husk (115.01–412.09 kPa) had a higher degree of flexural strength than MBCs made from sawdust (18.23–90.04 kPa). The highest flexural strength in this study was obtained from the corn husk MBCs containing 20% paper waste, demonstrating a statistically significant difference when compared to that of corn husk MBCs without paper waste (0%). In contrast, the sawdust MBCs with 40% paper waste exhibited the lowest flexural strength. The flexural strength of corn husk MBCs increased when 10% to 20% of paper waste was added but decreased once 30% to 40% of paper waste was added. Adding paper waste to sawdust MBCs did not improve their flexural strength at all as the flexural strength decreased from 90.04 when 0% paper waste was added to 18.23 kPa when 40% paper waste was added. The results of this investigation showed that adding paper waste into substrates at various levels caused the obtained MBC samples to exhibit varying flexural strengths. Particularly, the addition of 20% paper waste to corn husks improved flexural strength (412.09 kPa) by over 22%, surpassing that obtained by prior research that reported a flexural strength value of 320 kPa for MBCs made from corn husks and *L. sajor-caju* [28]. Normally, the flexural strength of MBCs is influenced by many parameters, such as the type of mycelium network, substrate types, and the pressing technique used [5,8,49]. In this study, the reduction of flexural strength values for MBCs produced from sawdust may be impacted by the incomplete mycelium coverage on the composite surface, which is an important factor for MBC bending strength (Figure 6B). At the same time, adding paper waste at a sufficient level into a composite material can improve the flexural strength of MBCs. This outcome is consistent with other research findings related to composite material studies that indicated that cellulose fiber reinforcement, particularly when it takes the form of recycled paper fibers, can function as reinforcement in composites [27,75,76]. This is because cellulose fibers provide added strength and elasticity, like more conventional reinforcing elements similar to fiberglass or carbon fibers [27]. Additionally, paper fibers that contain cellulose as an element, a natural polymer that can enhance the bonding between the fibers and the matrix material, with better adhesion, can lead to an improvement in flexural strength [77]. Moreover, the results of the flexural strength investigation were within the range of those reported by earlier studies on MBC materials: around 16.80–4400 kPa [8,40,48,59,60,73,78,79]. This is similar to many different types of synthetic foam, like polyimide (590–1360 kPa), polystyrene (70–700 kPa), polyurethane (200–5700 kPa), and phenolic formaldehyde resin foam (380–780 kPa) [28,49].

#### 3.2.3. Impact Strength

The obtained MBC samples for impact testing in this investigation are displayed in Figure 6C,D. The impact strength values of MBCs in this study ranged from 0.12 to 3.15 kJ/m^2^ (Figure 5C). MBCs produced from corn husk had a higher impact strength than MBCs made from sawdust for all substrate-to-paper waste ratios. Corn husk MBCs containing 30% paper waste had the highest impact strength (3.15 kJ/m^2^) while sawdust MBCs with 40% paper waste had the lowest impact strength (0.12 kJ/m^2^). Similar to the compression strength test results, as more paper waste was added into corn husk MBCs, the impact strength increased from 2.48 kJ/m^2^ with 0% paper waste to 3.15 kJ/m^2^ with 30% paper waste, indicating statistically significant differences. When another 10% of paper waste was added to the corn husk MBCs, the impact strength dropped to 1.61 kJ/m^2^. Moreover, adding paper waste to sawdust MBCs did not improve their impact strength; values decreased from 0.28 to 0.12 kJ/m^2^. This study found that the impact strength of the obtained MBC samples changed according to the different types of substrates and the amount of paper waste. MBCs produced from corn husk had an impact strength that was many times higher than MBCs produced from sawdust. This is because corn husks typically have longer and more fibrous cellulose structures compared to sawdust, which consists of small wood particles. Longer fibers can provide better reinforcement in the material, enhancing its ability to resist impact forces [80]. Additionally, adding paper waste to corn husks can also help to improve the impact resistance of MBCs. This is caused by the uniform distribution of paper waste fibers within the composite matrix, which can absorb and dissipate energy upon impact, hence reducing the force transmitted through the material and giving the composite stronger impact resistance [27,81,82]. Simultaneously, the decrease in impact strength observed in MBCs derived from sawdust may be associated with mycelial development issues, affecting bonding and substrate coverage.

Nevertheless, the outcomes of this study are consistent with those of many reports from earlier studies on composite materials, which indicated that the differences in impact strength of lignocellulosic composites primarily depend on many variables, including fiber and matrix strength, transference of load effectiveness, dissipation of fractures resistance, bonding capacity, fiber distribution, and structure [28,83,84]. The impact strength of MBCs is influenced by both the type of substrate used and the type of mycelium network [28]. When compared to the MBCs developed from prior research, it was found that most of the MBCs obtained in this study had increased impact resistance compared to previously reported MBCs [28]. Additionally, it was discovered that the obtained MBCs exhibited a similar impact resistance to wood particle boards (1–3.5 kJ/m^2^) [85] and paperboards (2–4 kJ/m^2^) [86], and also had a stronger impact resistance than many types of synthetic foam, like polyimide, polystyrene, polyurethane, phenolic formaldehyde resin foam, and polypropylene foam, with reported values of just around 0.1–1.63 kJ/m^2^ [28].

#### 3.2.4. Tensile Strength

Each MBC sample used for tensile testing in this study is shown in Figure 6E,F. The tensile strength of MBCs in this study ranged from 4.08 to 38.39 kPa (Figure 5D). The results showed that the tensile strength of MBCs made from corn husk had a higher degree of tensile strength than MBCs made from sawdust. However, adding paper waste to any of the MBCs did not improve the tensile strength at all. When paper waste was added to corn husk MBCs, the tensile strength decreased from 38.39 kPa with 0% paper waste to 29.60 kPa with 40% paper waste. Similarly, the tensile strength of sawdust MBCs decreased after adding more paper waste, as sawdust MBCs with 0% paper waste had a tensile strength of 10.35 kPa while sawdust MBCs with 40% paper waste had a tensile strength of 4.08 kPa. The findings of this study demonstrated that adding paper waste to the substrate may not be needed for improving the tensile strength of MBCs, which could potentially have negative impacts.

There are some reasons why adding paper waste might not improve tensile qualities. According to previous research, studies found that the tensile properties of MBCs are primarily influenced by mushroom mycelia used in the production affecting the mycelium binder network, the colonization within the substrate, along with the covering on the material surface, which varies according to the species of mushroom mycelium [5,40,49]. In general, MBCs frequently rely on a combination of mycelium, growth substrate, and other components for their greatest efficiency [87]. The addition of paper waste might disturb the delicate balance of mycelial development and delay colonization, which leads to problems regarding bonding and covering on the substrate [88]. At the same time, MBCs usually contain a mycelium-based matrix that provides a strong structure [89]. The addition of paper waste might affect mycelium matrix integrity, which might lead to weaker tensile properties.

### 3.3. Challenges and Future Perspectives

After an improvement in MBC properties having employed paper waste as a material mixture, it was found that the obtained MBCs displayed better qualities in many areas. Our current research demonstrated that the enhanced MBCs had similar properties to many kinds of traditional materials, like synthetic foams, some wood-based composite materials, along with paperboard products (Table 1), suggesting enormous possibilities for application as alternative replacement materials in the future. At the same time, some MBC properties need to be further studied and improved in order to address these drawbacks, reduce limitations on use, and unlock the full potential of MBCs. Addressing challenges and embracing future perspectives can lead to the creation of environmentally friendly, high-performance materials that can replace traditional options in home decorative items or non-load-bearing construction materials along with packaging applications (Figure 7A,B).

The development of the perfect MBCs that balance strength, durability, scalability, cost-effectiveness, acceptance, and environmental sustainability is a significant challenge. Due to multiple variables in the production process, such as mushroom mycelium strains, substrate types, growth conditions, along with manufacturing techniques used, it is still difficult to obtain consistent physical and mechanical properties in mycelium-based green composites. Thus, ensuring long-term durability, strength, and resistance to environmental factors like moisture and pests is crucial for home decorative items and packaging materials. Furthermore, determining cost-effective methods for large-scale production and making sure they maintain cost-competitiveness when compared to conventional materials is essential for scaling up production, while preserving quality and consistency. There is also a need to focus on finding new techniques and methods for large-scale production in order to obtain the perfect final product.

Future perspectives on MBC materials can be seen from many viewpoints and under a variety of terms. Firstly, in terms of environmentally friendly innovation, continued study could lead to more environmentally friendly and sustainable substitutes for conventional materials that can be used as new types of packaging material and home decorative items. Secondly, in terms of the circular economy, integrating a model where waste is continuously recycled could increase the viability of the industry. Thirdly, in terms of advanced material science, MBCs can still be developed to improve their mechanical and physical properties to expand their use into more varieties of industries. Moreover, in terms of market growth, when ecological consciousness increases, this may lead to a growth in the market for MBCs, opening opportunities for investment and innovation.

## 4. Conclusions

Improving MBC material properties through the addition of paper waste is a cost-effective and creative approach. This study explored MBCs produced from sawdust and corn husks with varying ratios of added paper waste, revealing that physical and mechanical properties vary based on substrate type and the amount of paper waste in the mixture. Regarding the physical properties of MBCs, adding paper waste into a mixture of both substrate types (corn husk and sawdust) was found to significantly increase material density. The highest values were observed at 274.51 kg/m^3^ in MBCs made from corn husks and 322.73 kg/m^3^ in MBCs made from sawdust with the addition of 40% paper waste. MBCs produced from corn husk with 10% paper waste (123.46%) absorb less water than MBCs without paper waste (132.44%), while the material swelling changes according to the degree of water absorption. Moreover, MBCs shrink more when more paper waste is added to the mixture, but there was no statistically significant difference between adding 10–30% paper waste and not adding any paper waste. Concerning mechanical properties, corn husk MBCs with 20% paper waste are stronger than MBCs without paper waste (compressive, flexural, and impact strengths), with the highest values recorded at 1314.75 kPa, 412.09 kPa, and 3.15 kJ/m^2^, respectively. At the same time, MBCs with paper waste (29.60–36.29 kPa in corn husk MBCs and 4.08–10.13 kPa in sawdust MBCs) have lower tensile strength than MBCs without paper waste (38.39 kPa in corn husk MBCs and 10.35 kPa in sawdust MBCs), which requires further investigation for improvement in the future. The combination of two-phase substrates and more than three phases in the production process is an interesting approach for future developments. Nevertheless, the developed MBCs had properties that were similar to those of many kinds of conventional materials, like synthetic foams (polystyrene and polyurethane), and some wood- and paper-based products (wood insulation boards, wood particle boards, and paper boards), making them a great substitution material in the fields of packaging materials and home decorative items. Future viewpoints on MBC materials in terms of modern material science, the circular economy, environmentally friendly innovation, and market growth may contribute to this material receiving more attention and becoming a real alternative material of the future because of its exceptional environmental friendliness and uniqueness.

## Figures and Tables

**Figure 1 polymers-16-00262-f001:**
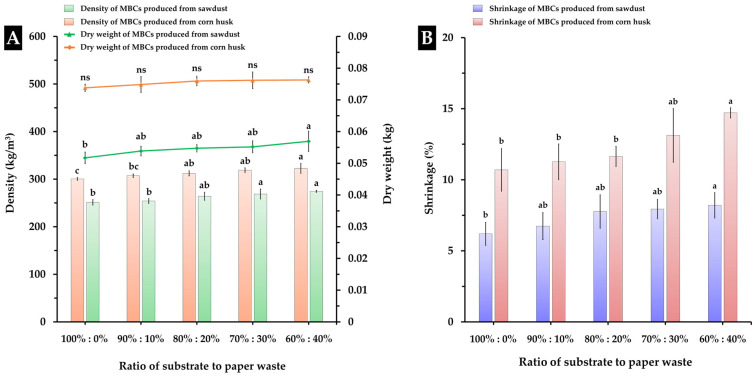
Density and dry weight (**A**), along with average shrinkage (**B**), of MBCs obtained from this study. The data are presented as means, and the error bars at each point indicate the ±standard deviation. Significant differences within the same experiment for each substrate type were determined by Duncan’s multiple range test, where distinct letters denote statistical significance (*p* ≤ 0.05).

**Figure 2 polymers-16-00262-f002:**
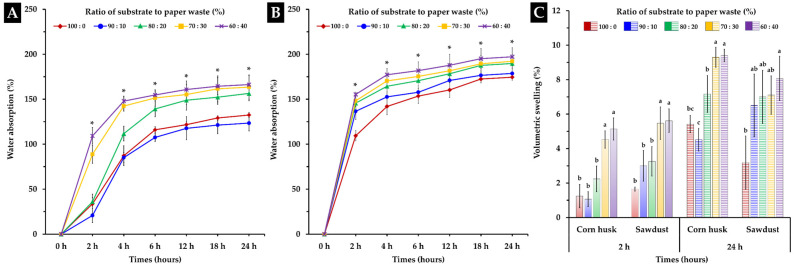
Water absorption behaviors of the MBCs produced from corn husk (**A**) and sawdust (**B**), along with volumetric swelling of the tested MBCs obtained in this study (**C**), are depicted. The data are presented as means, and the error bars at each point indicate the ±standard deviation. In (**A**,**B**), “*” denotes a significant difference based on Duncan’s multiple range test (*p* ≤ 0.05) at each point. Different letters in the experiment of each substrate type (**C**) denote significant differences according to Duncan’s multiple range test (*p* ≤ 0.05).

**Figure 3 polymers-16-00262-f003:**
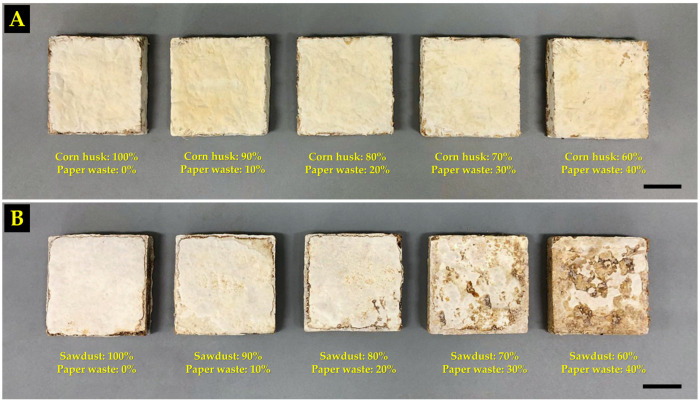
MBC samples for water absorption testing and volumetric swelling assessment were produced from corn husk (**A**) and sawdust (**B**), which were improved by paper waste. Scale bar = 2 cm.

**Figure 4 polymers-16-00262-f004:**
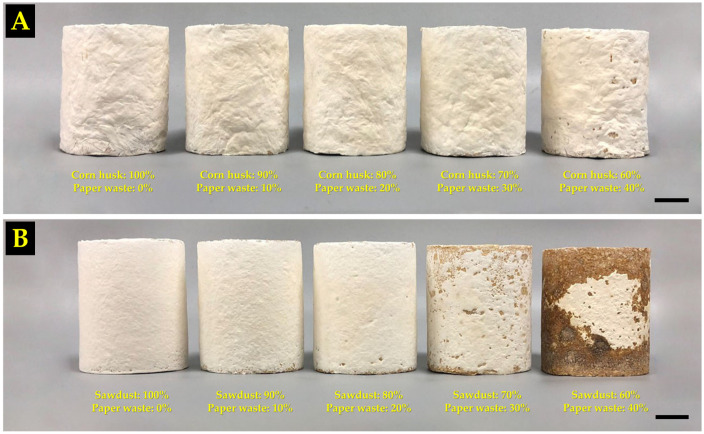
MBC samples for compression testing and density measurement were produced from corn husk (**A**) and sawdust (**B**), which were improved by paper waste. Scale bar = 2 cm.

**Figure 5 polymers-16-00262-f005:**
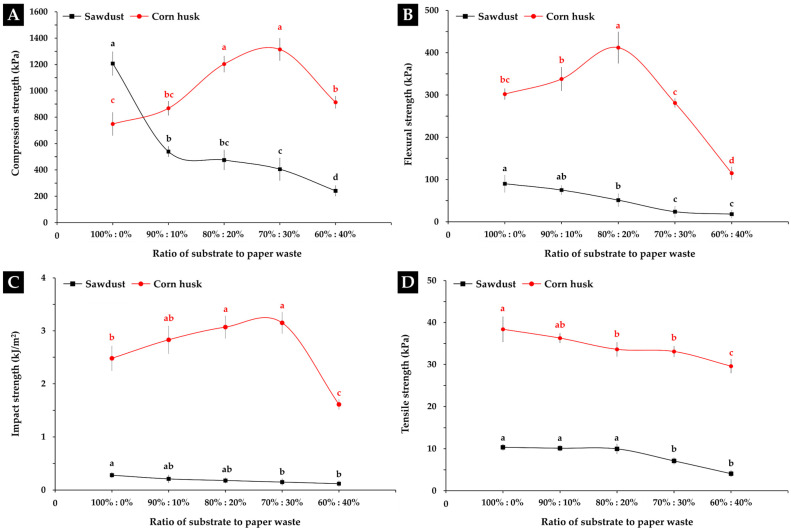
Compression (**A**), flexural (**B**), impact (**C**), and tensile strengths (**D**) of MBCs produced in this study. The data are presented as means, with error bars at each point indicating the ±standard deviation. Significant differences within the same experiment for each substrate type were determined by Duncan’s multiple range test, with distinct letters denoting statistical significance (*p* ≤ 0.05).

**Figure 6 polymers-16-00262-f006:**
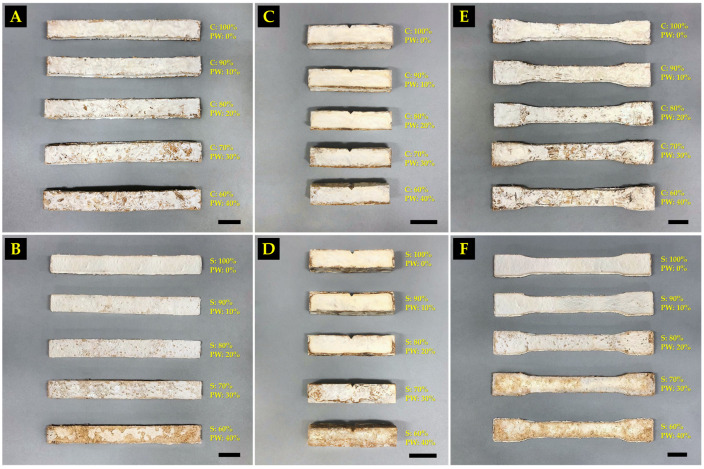
MBC samples produced from corn husk (**A**) and sawdust (**B**) for flexural testing. MBC samples produced from corn husk (**C**) and sawdust (**D**) for impact testing. MBC samples produced from corn husk (**E**) and sawdust (**F**) for tensile testing. Scale bar = 2 cm.

**Figure 7 polymers-16-00262-f007:**
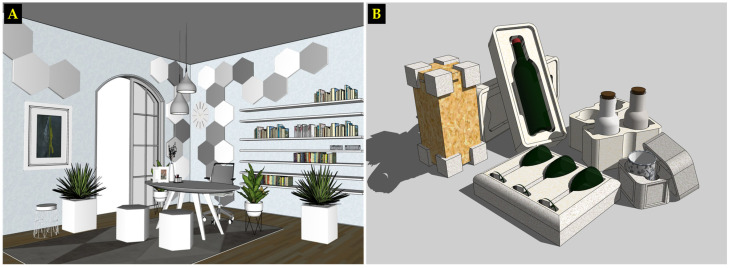
Possibilities of employing mycelium-based green composites in the future after improving their properties to create home decorative items (**A**) and packaging materials (**B**).

**Table 1 polymers-16-00262-t001:** Comparison of the MBC properties obtained in this study with those of traditional materials (modified from Aiduang et al. [28] and Jones et al. [49]).

Properties	MBCs in This Study	Polystyrene Foam	Polyurethane Foam	Paper Boards	Wood Insulation Board	Wood Particle Board
Density (kg/m^3^)	251.15 ± 5.67–322.73 ± 10.44	11–50	30–100	200–780	170–430	600–800
Shrinkage (%)	6.19 ± 0.81–14.71 ± 0.36	0.2–5	0.6–2	1–12	18.18–30.28	0.3–10
Water absorption (%)	123.46 ± 8.76–197.15 ± 10.10	0.03–9	0.01–72	300–350	55–380	30.1–200
Volumetric swelling (%)	4.51 ± 0.66–9.41 ± 0.36	-	-	5–12	1.89–5.25	15–25
Compression strength (kPa)	241.58 ± 39.19–1314.75 ± 111.45	30–690	2–48,000	40–10,000	100–1210	1800–3400
Flexural strength (kPa)	18.23 ± 9.25–412.09 ± 36.91	70–700	210–57,000	3760–4200	2000–2500	1500–7000
Impact strength (kJ/m^2^)	0.12 ± 0.05–3.15 ± 0.20	0.01–0.15	1.0–1.2	2–4	4.2–19	1–3.5
Tensile strength (kPa)	4.08 ± 0.72–38.39 ± 2.96	80–700	80–103,000	10–3340	350–1380	10,000–100,000

“-” = not reported [50,52,57,58,74,85,86,90,91,92,93,94].

## Data Availability

Data are contained within the article.

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
