# Peer review of "Improving the Physical and Mechanical Properties of Mycelium-Based Green Composites Using Paper Waste"

_polymers, 2024, doi:10.3390/polym16020262_

Round 1

Reviewer 1 Report

Comments and Suggestions for Authors

These studies make a significant contribution to the advancement of novel packaging materials. The mechanical tests are meticulously detailed, yet certain questions persist. Nonetheless, these inquiries do not diminish the significance of the accomplished research.

M&M

1. Please specify how was grinded rubber wood? 102-103

2. Please include the American Society for Testing and Materials (ASTM) standard number used. 123

3. Could you convert the measurements to the metric system (inches to meters) 144

4. Please add information about substrates humidity. 152

5.  You mentioned that after drying, the MBCs were stored in desiccators for further investigation.160

Could you clarify at what humidity the MBC samples were maintained during testing for moisture absorption?

Result

1. Can you explain why are highest density of this same composition of MBCs number not this same? 235- 239

2. Figure 1 requires improvement with additional explanations, and error bars with corresponding numbers need to be included or the letters explained.

3. Figure 2 needs enhancement, as A and B following the figure are unexplained; it remains unclear where corn husk and sawdust are positioned. Please provide explanations after the picture regarding the symbols (*) and letters representing error bars.

4. Figure 5 should be refined with additional explanations, and error bars with corresponding numbers must be added, or the letters clarified.

General question

1. Were your samples conditioned at a specific humidity level before mechanical tests?

2. I recommend excluding the data regarding moisture content before drying at 70°C, as the data on density and shrinkage are deemed sufficient. Additionally, in Section 2.3.3, you mention that water was added to achieve 60% moisture.

3. Could you explain how the mechanical and physical parameters change with mycelium growth longer than three day?

4. Can you elaborate on why you chose to use mushroom mycelium Lentinus sajor-caju?

5. Have you determined the thickness of Lentinus sajor-caju mycelium on the surface of the composite?

Author Response

The responses to our comments and suggestions can be found in the attached file.

Reviewer 2 Report

Comments and Suggestions for Authors

l230: why is it fundamental to determine the density of MBC according the substrate? Why does density of MBC need to be improved?

For all properties (compression, impact and flexural strength, water absorption and volumetric swelling, ...): why don't you take into account the standard deviations? By taking them into account some comments should be more moderate...

table 1: please indicate the standard deviations

Comments on the Quality of English Language

English well spelled

Author Response

(The authors gave the same response as above.)

Reviewer 3 Report

Comments and Suggestions for Authors

Dear authors,

The manuscript is prepared in a qualitative and research-appropriate manner.

The following comments increase the clarity of the results and help the readers to understand the presented results better.

1. Abstract need a more scientific approach rather supplying generic information. I suggests to enreach the abstract with real results for example - by what percentage did the mechanical strength increase by adding recycled paper, how did the water absorption change, etc.

2. In addition, the introduction should include the latest articles on mycelium composites, for example on basics of building with mycelium-based bio-composites, as well as characterization of self-growing biomaterials made of fungal mycelium and various lignocellulose-containing ingredients, where results on the use of sawdust and agricultural residue fiber in mycelial composites, the effect of chemical composition on mycelial growth, as well as mechanical properties and biodegradation were evaluated are discussed

3. Although the authors indicate that statistical analysis was used to evaluate the results, the statistical significance of the difference in results is not discussed in the discussion part. Please include it in the discussion of the manuscript and base the conclusions on statistical analysis.

​4. Since the authors base their results on different chemical composition in the samples (eg cellulose content), it would be desirable to determine at least the chemical composition of the starting materials (cellulose, hemicelluloses, lignin). In order to evaluate the use of nutrients during mycelial growth, determining the chemical composition of the final products would also provide an additional scientific aspect.

5. In order for the reader to more easily perceive the novelty of the research, it would be desirable to compare the author's results with the studies of other authors by comparing the absolute values (eg density and mechanical properties and water absorption capacity).

6. Recommendation for future research (not for this manuscript), since the authors emphasize that the material would be suitable for indoor design, studying the acoustic properties would be very useful and promising.

7. The conclusions are too general and descriptive, there is a lack of specific values for the analyzed properties.

Wishing you a successful year ahead and success in your studies

Best regards

Author Response

(The authors gave the same response as above.)

Reviewer 4 Report

Comments and Suggestions for Authors

Dear Authors

The researchers delve into the effects of adding paper waste to mycelium-based green composites (MBCs) produced from corn husks and sawdust. They discovered that incorporating paper waste can enhance the density, compressive strength, flexural strength, and impact strength of MBCs. Nevertheless, it also diminishes the tensile strength. The researchers conclude that paper waste can be a valuable addition to MBCs, recommending that future research further enhance the tensile strength of MBCs made with paper waste. The researchers comprehensively comprehend the existing research on MBCs and paper waste. They have effectively defined the research gap and their contribution to the field. The experimental design is sound and well-articulated. They have employed a combination of methods to obtain robust findings. The data analysis is thorough and transparent. They have presented their findings clearly and concisely. The discussion section is insightful and thought-provoking. They have concluded their research and recommended future research directions. Overall, the researchers have conducted a high-quality study that has the potential to impact the development of MBCs. Their research is well-written and easy to understand and will interest researchers and practitioners alike.

Some specific comments for considering:

1.       The Abstract could be improved by being more concise, specific, and engaging. It is vital to provide a clear and concise summary of the main goals and findings of the study, as well as to use strong verbs and active voice to make the abstract more engaging. Additionally, the Abstract should provide specific examples of the limitations of traditional materials, the benefits of using MBCs, and specific applications of MBCs with paper waste. The statement "This study aims to investigate the possible positive effects of utilizing paper waste as a reinforcing element to improve the physical and mechanical properties of MBC materials" is too long and could be shortened to "This study investigates the effects of using paper waste to improve MBC properties."

2.       The statement in the Abstract, "The outcomes demonstrate that the addition of paper waste into corn husk and sawdust significantly improves the density of the MBCs" could be simplified to "Adding paper waste increases the density of MBCs."

3.       The statement in the Abstract, "However, the addition of paper waste to sawdust does not have any influence on the MBC properties" could be simplified to "Adding paper waste to sawdust does not improve MBC properties."

4.       The statement in the Abstract "Some properties of MBCs might need to be further improved in the future to unlock their full potential" is too vague and could be more specific. What are the specific properties that need to be improved?

5.       Lines 57-60, in the statement, "Although MBCs are accepted as highly environmentally friendly materials, there are concerns related to their mechanical and physical properties that may limit their applications". The first part needs a literature source. There are accessible "Mycelium-Based Composite Materials Study of Acceptance" scientific documents (I would like to add that the wide acceptance of MBC is not so obvious. These materials are accepted, but not by everyone. This is due to the use of the fungus). Maybe the sentence "While MBCs are generally accepted [literature reference about acceptance] and lauded for their eco-friendliness, their mechanical and physical properties require further enhancement to realize their potential in various applications fully [7,8]" would be more precise.

6.       Linest 90-96. This paragraph does not fit the conclusion of the introduction; these are phrases typical of the end of an article. Please consider moving this paragraph to the end or deleting it.

7.       In my opinion, the conclusion of the manuscript does not effectively convey the laboratory experiment's main findings. The statement in the Conclusions section, "Adding 10% paper waste to corn husks can reduce the water absorption ability of MBCs" is not specific enough and could be simplified to "MBCs with 10% paper waste absorb less water than MBCs without paper waste." Additionally, it would be good to provide the values measured during the authors' experiments. Please also number the main conclusions from the research. It's better to read such summaries.

8.       The statement "Shrinkage value of the material increased with an increase in paper waste content" could be simplified to "MBCs shrink more when more paper waste is added to the mixture." As I proposed above, providing the values measured during the study would be beneficial.

9.       The statement "Adding 20% paper waste to corn husks was able to improve the compressive, flexural, and impact strengths of MBCs" could be simplified to "MBCs with 20% paper waste are stronger than MBCs without paper waste". Please provide the values measured.

10.   The statement "It was discovered that the tensile strength decreased when paper waste was added to the material mixture" could be simplified to "MBCs with paper waste have lower tensile strength than MBCs without paper waste." Please provide the values measured.

11.   The statement in the Conclusions section, "The developed MBCs had properties similar to many kinds of conventional materials," is too general and does not provide any specific examples of how MBCs can be used to replace conventional materials. Please provide the names of the conventional materials predisposed to replacing by MBCs

Generally, I consider the manuscript a valuable contribution to eco-friendly materials design. Please treat my comments as constructive suggestions for improving the article. Good luck in publishing (and many citations 😊)

Sincerely,

Author Response

(The authors gave the same response as above.)

Round 2

Reviewer 2 Report

Comments and Suggestions for Authors

All of my comments have been adressed

Author Response

Thank you for your response along with the valuable comments and suggestions provided to enhance our research.

Best regards

Reviewer 3 Report

Comments and Suggestions for Authors

Thank you for your responses and corrections, and improvements to the manuscript. In my opinion, the manuscript would still gain a significant overview if chemical analyses were performed on at least the starting materials, if not all the developed mycelial composites (this would also show changes in chemical structure and what fungi use as nutrients from lignocellulosic materials).

Author Response

(The authors gave the same response as above.)
